# Learning to Dive in Branch and Bound

**Max B. Paulus**
ETH Zurich
max.paulus@inf.ethz.ch

**Andreas Krause**
ETH Zurich
krausea@ethz.ch

## Abstract

Primal heuristics are important for solving mixed integer linear programs, because they find feasible solutions that facilitate branch and bound search. A prominent group of primal heuristics are diving heuristics. They iteratively modify and resolve linear programs to conduct a depth-first search from any node in the search tree. Existing divers rely on generic decision rules that fail to exploit structural commonality between similar problem instances that often arise in practice. Therefore, we propose *L2Dive* to learn specific diving heuristics with graph neural networks: We train generative models to predict variable assignments and leverage the duality of linear programs to make diving decisions based on the model's predictions. *L2Dive* is fully integrated into the open-source solver SCIP. We find that *L2Dive* outperforms standard divers to find better feasible solutions on a range of combinatorial optimization problems. For real-world applications from server load balancing and neural network verification, *L2Dive* improves the primal-dual integral by up to 7% (35%) on average over a tuned (default) solver baseline and reduces average solving time by 20% (29%).

## 1 Introduction

Mixed integer linear programming problems are optimization problems in which some decision variables represent indivisible choices and thus must assume integer values. They arise in numerous industrial applications, e.g., workload apportionment to balance server loads. They can be used to solve combinatorial problems or to verify the robustness of neural networks [11, 44]. We write a mixed integer linear program $M^\dagger := (c, P^\dagger)$, with $A \in \mathbb{R}^{m \times n}$, $b \in \mathbb{R}^m$, $c, x, \in \mathbb{R}^n$, $\underline{\pi}, \overline{\pi} \in \mathbb{R}^n_\infty$ and $\mathcal{I} \subseteq \{1, \dots, n\}$ indexing the variables restricted to be integrals, as

$$z^\dagger := \min_{x \in P^\dagger} c^\mathsf{T} x, \qquad P^\dagger = \{x \in \mathbb{R}^n \mid Ax = b, \ \underline{\pi} \le x \le \overline{\pi}, \ x_j \in \mathbb{Z} \ \forall j \in \mathcal{I}\} \qquad (1)$$

Typically, mixed integer linear programs are solved with a variant of branch and bound search [29, B&B]. This approach recursively builds a search tree, whose root represents the original problem in (1). Child nodes are created by introducing additional constraints (branching) to partition the set of feasible solutions. B&B uses bounds on the optimal solution to prune the tree and direct the search. To obtain strong upper bounds for $z^\dagger$, modern solvers typically rely on an array of primal heuristics. These are methods designed to quickly find good feasible solutions or an optimal solution $x^\dagger \in \mathcal{X} := \{x \in P^\dagger \mid c^\mathsf{T} x = z^\dagger\}$. Primal heuristics include problem-specific methods [e.g., 34, 30], variants of large neighborhood search [15, 13, 7, 38], rounding procedures [47, 4] or diving heuristics [see e.g., 6, 48].

Diving heuristics are a prominent group of primal heuristics. They are based on the linear program (LP) relaxation $M^* := (c, P^*)$ of the problem in (1) given by

$$z^* := \min_{x \in P^*} c^\mathsf{T} x, \qquad P^* = \{x \in \mathbb{R}^n \mid Ax = b, \ \underline{\pi} \le x \le \overline{\pi}\} \qquad (2)$$

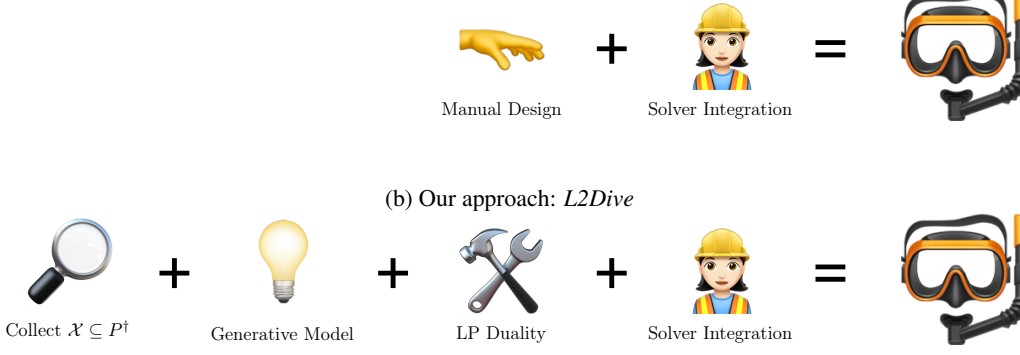

(a) Traditional approach: Diving heuristics in branch and bound

Manual Design Solver Integration

(b) Our approach: *L2Dive*

Collect $\mathcal{X} \subseteq P^\dagger$ Generative Model LP Duality Solver Integration

Figure 1: Traditional diving heuristics are based on generic manual heuristics and integrated efficiently into the branch and bound solver. In contrast, our approach *L2Dive* learns application-specific diving heuristics by collecting feasible solutions for a set of training instances to train a generative model. At test time, *L2Dive* uses the model predictions and leverages the duality of linear programs for diving. Finally, we integrate *L2Dive* into an open-source branch and bound solver.

Diving heuristics attempt to drive the LP solution $x^* \in \mathcal{X}^* := \arg\min c^\intercal x$ s.t. $x \in P^*$ towards integrality. For this purpose, they conduct a depth-first search from any node in the search tree by repeatedly modifying and resolving linear programs. Diving heuristics are popular, because linear programs can typically be solved relatively fast and hence a number of diving heuristics have been proposed. However, standard divers rely on generic rules that fail to exploit problem-specific characteristics. This is particularly severe in applications, where similar problem instances are solved repeatedly and structural commonality exists. In this setting, learning is a promising alternative to design effective divers with the ultimate goal of improving solver performance.

We propose *L2Dive* to learn such application-specific diving heuristics. *L2Dive* collects good feasible solutions for some instances of a particular application and trains a generative model to minimize a variational objective on them. The model is a graph neural network that, for a given mixed integer linear program, predicts an assignment of the integer variables. At test time, *L2Dive* uses the model prediction and leverages the duality of linear programs to select variables for diving and tighten their bounds. We fully integrate *L2Dive* into the open-source solver SCIP and demonstrate the effectiveness of our approach in two sets of experiments. Our approach is illustrated in First, we compare *L2Dive* against existing diving heuristics on a common benchmark of combinatorial optimization problems and show that it finds better feasible solutions with the same diving budget. Second, we use *L2Dive* within branch and bound and test it on two real-world applications where we find that *L2Dive* improves overall solver performance. For server load balancing, it improves the average primal-dual integral by up to 7% (35%) over a tuned (default) solver baseline and in neural network verification it reduces average solving time by 20% (29%).

## 2   Background

### 2.1   Diving Heuristics

Diving heuristics conduct a depth-first search in the branch and bound tree to explore a single root-leaf path. They iteratively modify and solve linear programs to find feasible solutions. Algorithm 1 illustrates a generic diving heuristic. A dive can be initiated from any node in the branch and bound tree and will return a (possibly empty) set of feasible solutions $\mathcal{X}$ for the original mixed integer linear program in (1). Typically, diving heuristics alternate between tightening the bound of a single candidate variable $x_j$ with $j \in \mathcal{C} \subseteq \mathcal{I}$ (line 5) and resolving the modified linear program (line 7), possibly after propagating domain changes [1]. The resultant solution may be integral ($x_j \in \mathbb{Z}, \forall j \in \mathcal{I}$) or admit an integral solution via rounding [line 9, 1]. Eventually, the procedure is guaranteed to result in at least one primal-feasible solution (line 10) or an infeasible program (line 6). However, in practice solvers may prematurely abort a dive to curb its computational costs,

for example by imposing a maximal diving depth (line 2), an iteration limit on the LP solver or a bound on the objective value.

Diving heuristics feature prominently in integer programming. In contrast to large neighborhood search, they only require repeatedly solving linear programs (in complexity class P) instead of small integer programs (NP-hard). As a result, they tend to be considerably faster and more applicable in practice. Diving heuristics may also leverage sophisticated rounding methods in each iteration which makes them more likely to be successful. Various diving heuristics have been proposed. We briefly list and describe the most common ones in Appendix A. They mostly differ in how they select variables for bound tightening (line 3). For example, *fractional* diving selects the variable with the lowest fractionality $|x_j^* - \lfloor x_j^* + 0.5 \rfloor|$ in the LP solution, while *linesearch* diving computes a score $s_j$ for each variable based on the LP solutions at the current and the root node. All of these diving heuristics choose integer variables for diving from the same set of candidates $\mathcal{C}$. This set of *divable* variables typically excludes integer variables that are *slack variables* [10, Chapter 1.1] or that have already been fixed ($\underline{\pi} = \overline{\pi}$). In general, diving heuristics apply to any mixed integer linear program. Empirically, however, standard divers often tend to be only effective for certain problems. For example, we found *Farkas* diving to deliver good results on a class of facility location problems, but to be ineffective for instances of maximum independent set (Section 5.1). As a result, modern solvers use an array of different diving heuristics during branch and bound. Selectively tuning this diving ensemble can be effective in improving solver performance on particular applications (Section 5). In contrast, our approach is to *directly learn problem-specific diving heuristics for particular applications* and demonstrate in our experiments that this yields improvements in solver performance over using a (tuned) ensemble of divers as is common practice.

---

**Algorithm 1:** Generic Diving Heuristic

**Input:** $M^\dagger$ with relaxation $M^*$, maximal depth $d_{\max}$
**Output:** $\mathcal{X}$, a (possibly empty) set of feasible solutions
**Require:** $s$, a scoring function to select variables for bound tightening

1  $d = 1$
2  **while** $d \leq d_{max}$ **do**
3      $j = \arg\max_{j \in \mathcal{C}} s_j$
4      Either $\underline{\pi}_j \leftarrow \lceil x_j^* \rceil$ or $\overline{\pi}_j \leftarrow \lfloor x_j^* \rfloor$
5      $P^* \leftarrow P^* \cap \{\underline{\pi}_j \leq x_j \leq \overline{\pi}_j\}$
6      **if** $P^*$ *is infeasible* **then** break;
7      $x^* = \arg\min\{c^\intercal x \mid x \in P^*\}$
8      **if** $x^*$ *is roundable* **then**
9         $\tilde{x} = \texttt{round}(x^*)$
10        $\mathcal{X} \leftarrow \mathcal{X} \cup \{\tilde{x}\}$
11     **end**
12     $d \leftarrow d + 1$
13     Possibly update candidates $\mathcal{C}$
14 **end**

---

## 2.2 Solver Performance and Primal Performance

The most intuitive way to assess the performance of a solver for mixed integer linear programs is by measuring *solving time*, i.e., the average time it takes to solve a set of instances. However, realistic instances are often not solved to completion, because this may be prohibitively expensive or a particular application only requires bounds on the optimal solution, which the solver readily yields. In these cases, it is common to consider the *primal-dual gap*[1]:

$$\gamma_{pd}(\tilde{z}, \check{z}^*) = \begin{cases} \frac{\tilde{z} - \check{z}^*}{\max(|\tilde{z}|, |\check{z}^*|)} & \text{if } 0 < \tilde{z}\check{z}^* < \infty \\ 1 & \text{else} \end{cases} \tag{3}$$

---

[1]Note that we use a definition of the primal-dual gap that is used in SCIP 7.0.2 for the computation of the integral and do not quantify the gap or the integral in per cent.

Here, $\tilde{z} := c^{\mathsf{T}}\tilde{x}$ is the upper (also known as primal) bound given by the feasible solution $\tilde{x}$ and $\check{z}^*$ is a global lower (also known as dual) bound on the optimal solution $z^\dagger$. Performance can be measured by solving instances with a fixed cutoff time $T$ and then computing the primal-dual gap $\gamma_{pd}(\tilde{z}_T, \check{z}_T^*)$, where $\check{z}_t^*$ and $\tilde{z}_t$ denote the solver's best lower and upper bounds at time $t$ (if non-existent, then $-\infty$ and $+\infty$ respectively).

**Primal-dual integral**   Unfortunately, measuring the primal-dual gap at time $T$ is susceptible to the particular choice of cutoff time. This is particularly troublesome, because the lower and upper bounds of branch and bound solvers tend to improve in a stepwise fashion. In order to alleviate this issue, it is common to integrate the primal-dual gap over the solving time and measure the primal-dual integral:

$$\Gamma_{pd}(T) = \int_{t=0}^{T} \gamma_{pd}(\tilde{z}_t, \check{z}_t^*)dt \tag{4}$$

**Primal gap**   When directly comparing diving heuristics with each other, it can be useful to consider primal performance instead of solver performance. Primal performance assesses the quality of the feasible solution $\tilde{x}$ a heuristic may find and can be measured by the *primal gap*:

$$\gamma_p(\tilde{z}) = \tilde{z} - z^\dagger \tag{5}$$

Sometimes, the primal gap is normalized by $|z^\dagger|$ which can be useful when $\gamma_p$ is averaged across disparate instances. We do not normalize $\gamma_p$ in this work.

## 3   Learning to Dive

We propose *L2Dive* to learn application-specific diving heuristics with graph neural networks. *L2Dive* uses a generative model to predict an assignment for the integer variables of a given mixed integer linear program. This model is learnt from a distribution of good feasible solutions collected initially for a set of training instances of a particular application. The model is a graph neural network closely related to the model in Gasse et al. [17]. It is trained to minimize a variational objective. At test time, *L2Dive* leverages insights from the duality of linear programs to select variables and tighten their bounds based on the model's predictions. We fully integrate *L2Dive* into the open-source solver SCIP.

### 3.1   Learning from feasible solutions

We propose to learn a generative model for good feasible solutions of a given instance $M^\dagger$. For this purpose, we first pose a conditional probability distribution over the variables $x$:

$$\log p_\tau(x|M^\dagger) :\propto \begin{cases} -c^{\mathsf{T}}x/\tau & \text{if } x \in \mathcal{X} \\ -\infty & \text{else} \end{cases} \tag{6}$$

The distribution $p_\tau(x|M^\dagger)$ is defined with respect to a set of good feasible solutions $\mathcal{X}$ for the given instance. Solutions with a better objective are given larger mass as regulated by the temperature $\tau$, while solutions that are not known to be feasible or are not good ($x \notin \mathcal{X}$) are given no probability mass. In practice, a model for diving will only need to make predictions on the *divable* variables $x_\mathcal{C}$ that are integral, non-fixed and not *slack* (Section 2). Hence, our model will target the marginal distribution $p_\tau^\mathcal{C}(x_\mathcal{C}|M^\dagger) := \sum_{\tilde{x}\in\mathcal{X}} \mathbb{1}\{x_\mathcal{C} \in \tilde{x}\}\, p_\tau(x|M^\dagger)$.

Our goal is to learn a generative model $q_\theta$ that closely approximates the distribution $p_\tau^\mathcal{C}$. The model $q_\theta$ will be used to make predictions on unseen test instances for which $p_\tau$ and $\mathcal{X}$ are unknown. To learn a good generative model, our objective is to minimize the Kullback-Leibler divergence between $p_\tau^\mathcal{C}$ and $q_\theta$,

$$\mathrm{KL}(p_\tau^\mathcal{C}||q_\theta) = \sum p_\tau^\mathcal{C}(x_\mathcal{C}|M^\dagger) \log \left( \frac{p_\tau^\mathcal{C}(x_\mathcal{C}|M^\dagger)}{q_\theta(x_\mathcal{C}|M^\dagger)} \right) \tag{7}$$

jointly over all training instances. The sum in (7) can be evaluated exactly, because the number of good feasible solutions in $\mathcal{X}$ tends to be small. Our model for $q_\theta$ is a variant of the graph neural

network described in Gasse et al. [17] and we give details in Appendix B. This model represents a mixed integer linear program as a bipartite graph of variables and constraints (Figure 3 in Appendix B) and its core are two bi-partite graph convolutions between variables and constraints (Figure 2 in Appendix B). Our variant uses some of the variable and constraint features from Paulus et al. [35] as well as batch normalization. It makes conditionally independent predictions for each integer variable, such that $q_\theta(x_\mathcal{C}|M^\dagger) := \prod_{j \in I} q_\theta(x_j|M^\dagger)$. For binary variables, $q_\theta(x_j|M^\dagger)$ is a Bernoulli distribution and the model outputs the mean parameter $\theta$. For general integer variables, $q_\theta(x_j|M^\dagger)$ is a sequence of Bernoulli distributions over the bitwise representation of the variable's integer domain and the model outputs a mean for each. Although conditionally independent predictions limit our model to unimodal distributions, this parameterization delivered strong empirical performance in our experiments (Section 5). It is possible to choose more delicate models for $q_\theta$, such as autoregressive models, but those will typically impose a larger cost for evaluations.

**Solution augmentation** At train time, we rely on the availability of a set of good feasible solutions $\mathcal{X}$ for each instance. This is required to define $p_\theta$ in (6) and evaluate the objective in (7) on all training instances. Several choices for $\mathcal{X}$ are possible, and the effectiveness of our approach may depend on them. For example, if $\mathcal{X}$ only contains poor feasible solutions, we cannot reasonably hope to learn any good integer variable assignments. The most obvious choice perhaps is to let $\mathcal{X} = \{x^\dagger\}$, where $x^\dagger$ is the best solution the solver finds within a given time limit $T$. However, solvers are typically configured to not only store $x^\dagger$, but also a set number of its predecessors. Thus, alternatively some or all of the solutions in store could be used to define $\mathcal{X}$ at no additional expense. Lastly, many instances of combinatorial optimization (e.g., set cover, independent set) exhibit strong symmetries and multiple solutions with the same objective value may exist as a result. These can be identified by using a standard solver to enumerate the solutions of an additional auxiliary mixed integer linear program as described in Appendix C. We used this method to augment solutions in $\mathcal{X}$ for some of our experiments. This technique may be of independent interest, because the problem of handling symmetries is ubiquitous in machine learning for combinatorial optimization [28]. While it can be expensive to collect feasible solutions for each training instance regardless of the choice of $\mathcal{X}$, this cost may be curbed, because we do not require $\mathcal{X}$ to contain the optimal solution. Moreover, the cost is incurred only once and ahead of training, such that all solver calls are embarrassingly parallelizable across instances. In some cases, a practitioner may be able to draw on previous solver logs and not be required to expend additional budget for data collection. Finally, any training expense will be ultimately amortized in test time service.

### 3.2 Using a generative model for diving

At test time, we use our generative model $q_\theta$ to predict an assignment for the *divable* integer variables $x_\mathcal{C}$ of a given instance. Typically, we will choose $\hat{x}_\mathcal{C} = \arg\max q_\theta(x_\mathcal{C}|M^\dagger)$ to predict an assignment, but assignments can also be sampled from the model, if multiple dives are attempted in parallel. To use the prediction for diving, we need to decide which variable to select (line 3 in Algorithm 1) and how to tighten its bounds (line 4). Ideally, our decision rules will admit a feasible solution at shallow depths, i.e., only a few bounds must be tightened to result in an integral or roundable solution to the diving linear program. Which variables should we tighten for this purpose and how? Compellingly, the theory of linear programming affords some insight:

**Proposition 1.** *Let $\tilde{x}$ be a feasible solution for $M^\dagger$ as in (1). For the linear program $M^*$, its dual linear program $M_D^*$ is defined in (11) in Appendix D. Let $y^* := (y_b^*, y_{\underline{\pi}}^*, y_{\overline{\pi}}^*)$ be an optimal solution for $M_D^*$. Let $\underline{J}(\tilde{x})$ and $\overline{J}(\tilde{x})$ index the set of variables that violate complementary slackness (12) between $\tilde{x}$ and $y^*$, such that*

$$\underline{J}(\tilde{x}) := \{j \mid (\tilde{x}_j - \underline{\pi}_j)\, y_{\underline{\pi}\,j}^* > 0\}$$
$$\overline{J}(\tilde{x}) := \{j \mid (\tilde{x}_j - \overline{\pi}_j)\, y_{\overline{\pi}\,j}^* > 0\}$$

*and define $J(\tilde{x}) := \underline{J}(\tilde{x}) \cup \overline{J}(\tilde{x})$. Let $M_J^* := (c, P_J^*)$ be the linear program, where the bounds of all variables indexed by $J(\tilde{x})$ are tightened, such that*

$$P_J^* = P^* \cap \{x \in \mathbb{R}^n \mid x_j \geq \tilde{x}_j \ \forall j \in \underline{J}(\tilde{x}), \ x_j \leq \tilde{x}_j \ \forall j \in \overline{J}(\tilde{x})\}$$

*Then, $\tilde{x}$ is an optimal solution to the linear program $M_J^*$, i.e., $\tilde{x} \in \arg\min_{x \in P_J^*} c^\mathsf{T} x$.*

*Proof.* $\tilde{x}$ is clearly a feasible solution for $M_J^*$. $y^*$ is a feasible solution for the dual linear program of $M_J^*$, because it is feasible for $M_D^*$. $\tilde{x}$ and $y^*$ satisfy complementary slackness, hence $\tilde{x}$ is optimal. $\quad\square$

This suggests that for a prediction $\hat{x}_\mathcal{C}$, the bounds of variables in $J(\hat{x}_\mathcal{C})$ should be tightened to restore complementary slackness. If the integer variable assignment $\hat{x}_\mathcal{C}$ is feasible and the candidate set includes all integer variables, this will yield a diving linear program for which the assignment is optimal and that may be detected by the LP solver. Unfortunately, this is not guaranteed in the presence of *slack* variables (where typically $\mathcal{C} \subset \mathcal{I}$) or if multiple optimal solutions exist (some of which may not be integer feasible). In practice, it may thus be necessary to tighten additional variables in $\mathcal{C}$. Inspired by Proposition 1, we propose the following *dual reasoning* rule to select to select a variable $j^* \in \mathcal{C}$ for tightening

$$j^* = \arg\max_{j \in \mathcal{C}} s_j := q_\theta(\hat{x}_j) + \mathbb{1}\{j \in J(\hat{x}_\mathcal{C})\} \tag{8}$$

This rule will select any variables in $J(\hat{x}_\mathcal{C})$ before considering other variables for tightening. The score $s$ breaks ties by selecting the variable in whose predictions the model is most confident in. Conveniently, the set $J(\hat{x})$ can be easily computed on the fly from the dual values $y^*$, which standard LP solvers readily emit on every call at no additional expense. We tighten $\underline{\pi}_j = \hat{x}_j$ if $\hat{x}_j \leq x_j^*$ and we tighten $\overline{\pi}_j = \hat{x}_j$ if $\hat{x}_j \geq x_j^*$ to replace line 4 in Algorithm 1. We update the candidate set $\mathcal{C}$ in line 13 to exclude variables whose value has been fixed, i.e., $\underline{\pi}_j = \overline{\pi}_j$ as usual. We validate dual reasoning in an ablation study in section 5.1.

### 3.3 Deployment

We fully integrate *L2Dive* into the open-source solver SCIP 7.0.2 [16]. This solver exposes a plug-in for diving heuristics that implements an optimized version of Algorithm 1 in the programming language C. We extend the solver's Python interface [32] to include this plug-in and use it to realize *L2Dive*. This integration facilitates direct comparison to all standard divers implemented in SCIP (Section 5.1) and makes it easy to include *L2Dive* in SCIP for use in branch and bound (Section 5.2). Importantly, we call our generative model only once at the initiation of a dive to predict a variable assignment. While predictions may potentially improve with additional calls at deeper depths, this limits the in-service overhead of our method. It also simplifies the collection of training data and produced good results in our experiments (Section 5).

## 4 Related Work

Nair et al. [33] propose a method that learns to tighten a subset of the variable bounds. It spawns a smaller sub-integer program which is then solved with an off-the-shelf branch and bound solver to find feasible solutions for the original program. Sonnerat et al. [41] improve this approach using imitation learning. Others explore reinforcement learning [49] or hybrids [40], but only focus on improving primal performance. All of these methods are variants of large neighborhood search [39, 2, 36], where a neighborhood for local search is not proposed heuristically, but learnt instead. In contrast, our approach *L2Dive* does not propose a fixed neighborhood and it does not require access to a branch and bound solver to run. Instead, we use our model's predictions to iteratively modify and solve linear programs instead of sub-integer programs. In practice, linear programs tend to solve significantly faster which makes *L2Dive* more applicable. Khalil et al. [26] propose a method to learn variable assignments from good feasible solutions, but combine their model predictions with a heuristic rule for node selection, whereas we consider diving.

Overall, there is vivid interest in exploring the use of machine learning for integer programming [5, 52, 31]. With regard to branch and bound, several works learn models for variable selection in branching [25, 3, 17, 19, 42, 51]. Others focus on node selection in the search tree [20, 50] or deal with cutting plane management [35, 22, 43, 9, 45]. Further, related work includes both general [23, 24, 46] and specific [12, 9, 8] attempts of learning to configure the solver. To the best of our knowledge, we are the first to propose *learning to dive* to improve the performance of branch and bound solvers.

# 5 Experiments

The goal of our work is to learn application-specific diving heuristics to improve on existing diving heuristics. We view other primal methods (Section 4) as complementary, and accordingly compare primarily to other diving heuristics. We evaluated the effectiveness of *L2Dive* in two different experiments and on a total of six datasets. The first set of experiments (Section 5.1) was designed to study the diving performance of *L2Dive* in isolation and compare it against existing diving heuristics. On a benchmark of four synthetic combinatorial optimization problems from previous work [17], we performed single dives with each diver and measured the average primal gap. We found that *L2Dive* outperformed all existing divers on every dataset and produced the best solutions amongst all divers. The second set of experiments (Section 5.2) directly included *L2Dive* into the branch and bound process of the open-source solver SCIP. The solver called *L2Dive* in place of existing diving heuristics and our goal was to improve overall performance on real-world mixed integer linear programs. We considered instances from neural network verification [33] and server load balancing in distributed computing [18]. We measured performance with *L2Dive* against the default configuration and a highly challenging baseline that tuned the solver's diving ensemble. We found that *L2Dive* improved the average primal-dual integral by 7% (35%) on load balancing and improved average solving time by 20% (29%) on neural network verification over the tuned (default) solver.

We collected data for training and validation: In all experiments, we extracted a bipartite graph input representation of each instance's root node. On all but two datasets, we chose $\mathcal{X} = \{x^\dagger\}$ where $x^\dagger$ is the best solution the solver finds within a given time limit $T$. For set cover and maximum independent set only, we observed strong symmetries and used the solution augmentation described in Appendix C. We trained separate models for each dataset. We trained each model with ADAM [27] for 100 epochs in the first set of experiments and for 10 epochs in the second set of experiments. We individually tuned the learning rate from a grid of $[10^{-2}, 10^{-3}, 10^{-4}]$. For each dataset, we chose the batch size to exhaust the memory of a single NVIDIA GeForce GTX 1080 Ti device. We validated after every epoch and chose the model that achieved the best validation loss. In all experiments, we use the mode prediction of the generative model and only perform a single dive from a given node. We do not attempt multiple dives in parallel and did not use any accelerators at test time. In all experiments, we only call *L2Dive*'s generative model once at the beginning of a dive to limit the in-service overhead from serving the graph neural network.

## 5.1 Diving with *L2Dive*

With this first set of experiments, we studied the diving performance of *L2Dive* and compared it against existing diving heuristics. We used the same benchmark as previous work [17]. This benchmark consists of four different classes of combinatorial optimization problems, including set covering, combinatorial auctions, capacitated facility location and maximum independent sets. For each class, we used 2000 instances in total; we trained on 1000 instances and validated and tested on 500 instances respectively. We presolved all instances before diving, but did not branch and disabled cutting planes and other primal heuristics as our interest is solely in diving. We compared *L2Dive* against all other standard diving heuristics that are implemented in the open-source solver SCIP and do not require an incumbent solution. This includes *coefficient*, *fractional*, *linesearch*, *pseudocost*, *distributional*, *vectorlength* [6] and *Farkas* diving [48]. We briefly describe these baseline divers in Appendix A. In addition, we considered three trivial divers that respectively fix integer variables to their lower (*lower*) or upper limit (*upper*) or either with equal probability (*random*). All divers ran with the same diving budget ($d_{max} = 100$) and their execution was directly triggered by the user after resolving the root node. We ignore the few test instance that were solved directly at root by SCIP before we could initiate a dive.

*L2Dive* outperformed all other standard divers (Table 1). It achieved the lowest average test primal gap on each of the four problem classes. The improvements over the *best heuristic* diver ranged from roughly 15% for combinatorial auctions to more than 70% for independent sets. The trivial divers only found solutions that are significantly worse, which indicates that *L2Dive* learnt to exploit more subtle patterns in the problem instances to find better feasible solutions. Some baseline divers (e.g., *linesearch*, *distributional*) failed to consistently outperform the trivial divers across all problem classes and *best heuristic* diver varied (*pseudocost* diving for combinatorial auctions, *Farkas* diving for facility location, *vectorlength* diving for set cover, independent set). This confirms that in practice most diving heuristics tend to be specialized and work particularly well for particular problem classes.

Table 1: *L2Dive* finds better feasible solution on all four problem classes than existing diving heuristics. Average primal gap with standard error on test set. Best diver is in **bold** and best heuristic is in *italics*.

|  | SET COVER | COMB. AUCTION | FAC. LOCATION | IND. SET |
|---|---|---|---|---|
| *L2Dive* | **55** (3) | **222** (7) | **160** (10) | **5** (1) |
| *Best heuristic* | *95* (3) | *256* (8) | *484* (7) | *18* (2) |
| *Coefficient* | 3,700 (55) | 671 (11) | 762 (9) | 246 (4) |
| *Distributional* | 3,900 (50) | 1,504 (12) | 760 (9) | 196 (3) |
| *Farkas* | 105 (3) | 476 (9) | *484* (7) | - |
| *Fractional* | 3,726 (57) | 672 (10) | 1,058 (11) | 232 (4) |
| *Linesearch* | 1,269 (24) | 467 (10) | 1,036 (15) | 77 (1) |
| *Pseudocost* | 195 (9) | *256* (8) | 505 (11) | 32 (2) |
| *Vectorlength* | *95* (3) | 832 (20) | 840 (19) | *18* (1) |
| *Random* | 416 (13) | 704 (12) | 902 (14) | 78 (2) |
| *Lower* | 2,918 (63) | 1,587 (11) | 623 (8) | 171 (5) |
| *Upper* | 239 (6) | 611 (11) | 828 (14) | 62 (2) |

*L2Dive* is a generic recipe to design effective divers for any specific application. Finally, we found that the mode predictions of our learnt models were rarely feasible (e.g., set cover, combinatorial auctions) or yielded poor solutions (e.g., independent set). This highlights that learning a generative model for diving may be a more promising approach than trying to predict feasible solutions directly.

In order to validate the dual reasoning rule proposed in subsection 3.2, we paired *L2Dive* with two alternative rules for variable selection in capacitated facility location. The first ablative rule chooses a variable $j \in \mathcal{C}$ uniformly at random, i.e., $s_j^{rand} \sim \mathcal{U}_{[0,1]}$. The second ablative rule simply uses model confidence, i.e., $s_j^{conf} = q_\theta(\hat{x}_j)$ and unlike dual reasoning does not treat variables $j \in J(\hat{x}_{\mathcal{C}})$ preferentially. We found that even when variables were selected uniformly at random (where the model prediction is used only for bound tightening), *L2Dive* outperformed the best standard diver (Table 2). However, selecting variables whose model predictions are more certain significantly improved performance by a large margin, while additionally employing dual reasoning tended to improve performance further for capacitated facility location. The effectiveness of dual reasoning is likely problem-specific, as dual reasoning will collapse to the model confidence rule, if the set $J \cap \mathcal{C}$ is empty. To test the generalization performance of *L2Dive* to larger instances, we performed an additional ablation study and report the results in Appendix E.

Table 2: Dual reasoning in *L2Dive* tends to improve diving performance on capacitated facility location. Even selecting variables for diving uniformly at random outperforms the best heuristic diver, but using model confidence (and dual reasoning) facilitates significant improvements.

|  | FAC. LOCATION |
|---|---|
| *L2Dive* | **160** (10) |
| *L2Dive* (with $s_j^{conf}$) | **164** (10) |
| *L2Dive* (with $s_j^{rand}$) | **335** (14) |
| *Best heuristic* | *484* (7) |

## 5.2 *L2Dive* in branch and bound

With this second set of experiments, our goal was to use *L2Dive* within branch and bound to improve solver performance on real-world mixed integer linear programs. To this end, we included *L2Dive* into the open-source solver SCIP. We disabled all other diving heuristics in SCIP and dive with *L2Dive* from the root node. We found this to work well, but results for *L2Dive* may likely improve with a more subtle schedule for *L2Dive* or by integrating *L2Dive* into the solver's diving ensemble.

We considered two strongly contrasting applications from previous work. The first application deals with safely balancing workloads across servers in a large-scale distributed compute cluster. This problem is an instance of bin-packing with apportionment and can be formulated as a mixed integer linear program. We used the dataset from Gasse et al. [18] which contains 9900 instances for training and 100 instances for validation and testing respectively. Solving these instances to optimality is prohibitively hard[2] and we therefore set a time limit of $T_{\text{limit}} = 900$ seconds, both for data collection and test time evaluations. The second application deals with verifying the robustness of neural networks. This problem can be formulated as a mixed integer linear program [11, 44]. We considered the instances from Nair et al. [33], but used the same subset as Paulus et al. [35] who disregard trivial and numerically unstable instances. This dataset contains 2384 instances for training, 519 instances for validation and 545 instances for testing. These instances are challenging, but can mostly be solved within a reasonable time. We set a limit of $T_{\text{limit}} = 3600$ seconds, both for data collection and test time evaluations.

To assess solver performance, we measure solving time $T$ for neural network verification and the primal-dual integral $\Gamma_{pd}(T_{\text{limit}})$ for server load balancing. Both measures fully account for the entire in-service overhead of *L2Dive* (e.g., computing the bipartite graph representation from the tree node, forward-propagating the generative model, diving etc.), because the *L2Dive* diver is directly included into SCIP and called by the solver during the branch and bound process. Our experiments were conducted on a shared distributed compute cluster. To reduce measurement variability, we ran all test time evaluations repeatedly on machines equipped with the same Intel Xeon Gold 5118 CPU 2.3 GHz processors for three different seedings of the solver. We batched evaluations randomly across instances and methods to be processed sequentially on the same machine. We report test set means and standard errors over the three different random seeds.

Table 3: *L2Dive* improves the performance of the branch and bound solver SCIP on real-world applications. When using *L2Dive* instead of standard divers, the average primal-dual integral for load balancing improves by 7% (35%) and solving time on neural network verification shrinks by 20% (29%) against the tuned (default) solver.

| | LOAD BALANCING | | NEURAL NETWORK VERIF. | |
|---|---|---|---|---|
| | Primal-dual Integral | Wins | Solving Time | Wins |
| SCIP | | | | |
| *Default* | 4,407 (34) | 0 (0) | 55.8 (2.3) | 54 (5) |
| *No diving* | 4,221 (21) | 0 (0) | 53.7 (0.6) | 40 (4) |
| *Tuned* | 3,067 (10) | 7 (3) | 49.9 (2.8) | 164 (3) |
| *L2Dive* | **2,863 (13)** | **93 (3)** | **39.8 (2.3)** | **287 (5)** |

*L2Dive* improved solver performance ad-hoc (Table 3, *L2Dive*). On load balancing, *L2Dive* improved the average primal-dual integral by over 30% from the solver at default settings (Table 3, *Default*). On neural network verification, *L2Dive* reduced the average solving time from approximately 56 seconds to less than 40 seconds (35%). As a control, we also ran SCIP without any diving and surprisingly found small improvements on both datasets (Table 3, *No diving*). The solver's default setting are calibrated on a general purpose set of mixed integer programs and are typically a challenging baseline to beat. However, our results suggests that SCIP's divers are either ineffective or may be poorly calibrated for these two applications. For this reason, we decided to tune the diving heuristics of the solver to seek an even more challenging baseline for comparison. We leveraged expert knowledge and random search to find strong diving ensembles in the the vicinity of the default configuration. Then, we selected the best tuned solver configuration on a validation set using the same budget of solver calls that *L2Dive* expended for data collection. Details are in Appendix F. Our tuned solver baseline (Table 3, *Tuned*) significantly improved performance over *Default*, but was still outperformed by *L2Dive*. This highlights that our approach to learn specific divers may be more promising than fitting ensembles of generic diving heuristics to a particular application. Overall, *L2Dive* achieved the best average performance on 93 (out of 100) test instances for load balancing, and achieved the best

---

[2]Using a Xeon Gold 5118 CPU processor with 2.3 GHz and 8 GB of RAM, none of the instances could be solved with SCIP 7.0.2 at default settings within an hour.

average performance on 287 (out of 545) test instances for neural network verification, more than the three SCIP configurations collectively.

## 6 Conclusions

We presented *L2Dive* to learn application-specific diving heuristics for branch and bound. Our approach combines ideas from generative modeling and relational learning with a profound understanding of integer programs and their solvers. We tested *L2Dive* on a range of applications including combinatorial optimization problems, workload apportionment and neural network verification. It found better feasible solutions than existing diving heuristics and facilitated improvements in overall solver performance. We view our work as yet another example that demonstrates the fruitful symbiosis of learning and search to design powerful algorithms.

## Broader Impact and Limitations

This work presents a method to use machine learning for improving the performance of branch and bound solvers. Branch and bound is a powerful general-purpose method for solving mixed integer linear programs which appear frequently across business, science and technology. Therefore, we expect the impact of this work to be overwhelmingly beneficial. However, in cases where integer programming is exploited with ill intentions, our work may potentially have a harmful societal impact.

There are limitations in learning diving heuristics for specific applications. For example, in some cases the set of training instances may be small or collecting feasible solutions could be prohibitively expensive. In such cases, it may be desirable to transfer models from other applications or to utilize self-supervised representations that require fewer labelled examples for training [14]. This is a natural direction to explore in the future, for this and other work at the intersection of machine learning and integer programming. Alternatively, one may attempt to learn a universal diving heuristic using a diverse set of instances from a variety of applications. However, the extent to which machine learning can prove effective in this setting, for diving or other sub-routines, remains an open question.

## Acknowledgements

MBP gratefully acknowledges support from the Max Planck ETH Center for Learning Systems. Resources used in preparing this research were provided, in part, by the Sustainable Chemical Processes through Catalysis (Suchcat) National Center of Competence in Research (NCCR).

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

## A  Diving Heuristics

In Table 4, we briefly describe existing diving heuristics. In addition, SCIP's diving ensemble includes *adaptive* diving, *conflict* diving, *objective pseudocost* diving and *guided* diving. We did not compare against these heuristics in Section 5.1, because they either choose from the other heuristics (*adaptive*), were ineffective (*conflict*), require a feasible solution (*guided*) or do not use the generic diving algorithm (*objective pseudocost*). However, these divers are active for the baselines in Section 5.2. Berthold [6] gives a more detailed illustration of some of the divers.

Table 4: Overview of standard diving heuristics.

| Diver | Description |
|---|---|
| *Coefficient* | Selects the variable with the minimal number of (positive) up-locks or down-locks and bounds it in the corresponding direction; ties are broken using *Fractional* diving. |
| *Distribution* | Selects a variable based on the solution density following [37]. |
| *Farkas* | Attempts to construct a Farkas proof, *Farkas* diving bounds a variable in the direction that improves the objective value and selects the variable for which the improvement in the objective is largest. |
| *Fractional* | Selects the variable with the lowest fractionality $\lvert x_j^* - \lfloor x_j^* + 0.5 \rfloor \rvert$ in the current LP solution $x^*$ and bounds it in the corresponding direction. |
| *Linesearch* | Considers the ray originating at the LP solution of the root and passing through the current node's LP solution $x^*$, selects the variable $j$ whose coordinate hyperplane $x_j = \lfloor x_j^* \rfloor$ or $x_j = \lceil x_j^* \rceil$ is first intersected by the ray. |
| *Pseudocost* | Selects and bounds a variable based on its branching pseudocosts, its fractionality and the LP solutions at the root and the current node. |
| *Vectorlength* | Inspired by set partition constraints, selects a variable for which the quotient between the objective cost from bounding it and the number of constraint it appears in is smallest. |

## B  Bipartite Graph and Graph Neural Network for *L2Dive*

### B.1  Bipartite Graph

In our experiments, we represent a mixed integer linear program as a bipartite graph of variables and constraints. This idea is taken from Gasse et al. [17] and has been adopted by many others since. Each variable $x_j$ and each constraint $A_{i\cdot}$ in the program gives rise to a node in a bipartite graph. For every non-zero coefficient $A_{ij}$ in a linear constraint of the program the nodes of the corresponding variable $j$ and constraint $i$ are joined by an undirected edge. Each node is associated with a vector of features. We use the same features as Paulus et al. [35] but exclude cut-specific ones, such that variable nodes use 36 features and constraint nodes use 69 features in total.

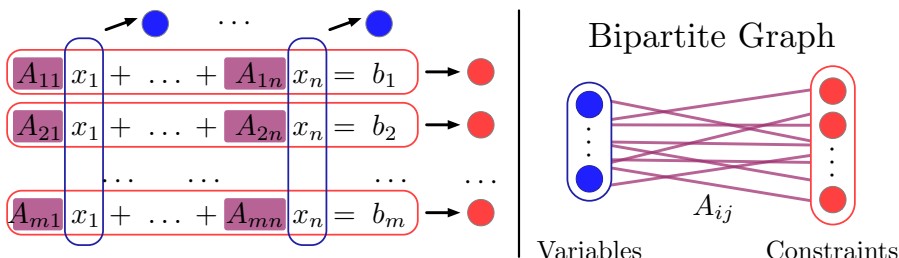

Figure 2: Our model uses a bipartite graph representation to represent a mixed integer linear program as in [17]. Both variables and constraints of the program represent nodes in a bipartite graph. Edges in the graph correspond to non-zero coefficients in the linear constraint of the program. We use the same features for variables and constraints as in [35], but exclude cut-specific ones.

## B.2   Graph Neural Network

Our model is a graph neural network that closely resembles the model proposed in Gasse et al. [17] and uses some of the modifications from Paulus et al. [35]. The model is sketched in Figure 2. After separately applying batch normalization to both variable and constraint nodes, we embed all nodes in a 64-dimensional space using a multi-layer perceptron with a single hidden layer. This is followed by two bipartite graph convolutions, first from variables to constraints and then from constraints to variables, as in Gasse et al. [17]. Finally, we predict from the convolved variable embedding the parameters $\theta$ of a probability distribution $q_\theta(x_j|M^\dagger)$ for each *divable* integer variable using another multi-layer perceptron with a single hidden layer. Binary variables are the most common variables in integer programs, and all the instances we considered feature exclusively binary *divable* variables. For binary variables, $q_\theta(x_j|M^\dagger)$ is a Bernoulli distribution and the model outputs the mean parameter $\theta$, such that $\mathbb{P}(x_j = 1) = \theta$. For general integer variables, we suggest to consider the bitwise representation of the integer domain. For example, the domain of a variable that can assume no more than eight unique integer values can be represented using at most four bits. Each of these bits can be parameterized with its own Bernoulli distribution and the model outputs a mean for each. This approach may be more favorable for diving than for other applications, because *slack variables* from cutting plane constraints (whose domains are not known initially and may be large) are not *divable*. In cases where outputting a fixed-size array of Bernoulli parameters may not be applicable, a variable length array could be outputted by using a recurrent layer, such as an LSTM [21], as is proposed in Nair et al. [33]. Alternatively, a fixed size array could still be used with an additional bit to indicate integer overflow that must be handled appropriately. For example in diving, variables for which the model predicts overflow may be ignored. Further, the frequency with which overflow would be encountered in practice could plausibly be reduced by making predictions relative to the current solution $\lfloor x_j^* \rfloor$ rather than with respect to the lower variable bound $\underline{\pi}_j$.

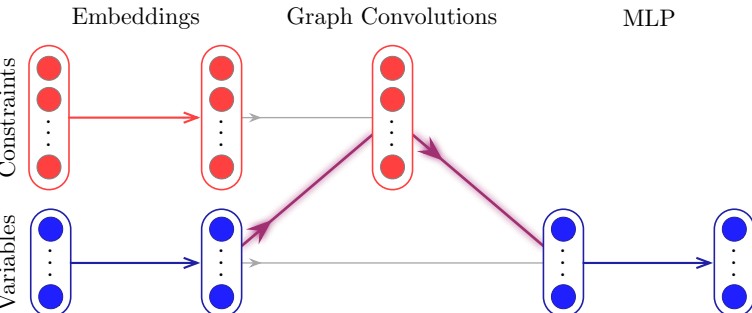

Figure 3: Our model is a graph neural network based on Gasse et al. [17] and Paulus et al. [35]. It first embeds variable and constraint nodes using batch normalization and a single-layer feedforward network. It then convolves variable and constraint features with two bi-partite graph convolutions as in Gasse et al. [17]. Finally, for each variable it outputs the parameters of the distribution $q_\theta(x_j)$ using another single-layer feedforward network.

## C   Solution Augmentation by Counting Optimal Solutions

Many integer and mixed integer linear programs exhibit strong symmetries, particularly those from combinatorial optimization. In these cases multiple optimal solution may exist, and in particular different integer variable assignments that correspond to an optimal solution. It is possible to identify those by first solving for $z^\dagger$ to then define the auxiliary mixed integer linear program,

$$\min_{x \in P^\mathcal{X}} c^\intercal x, \qquad P^\mathcal{X} = \{x \in P^\dagger, c^\intercal x = z^\dagger\} \tag{9}$$

Standard solvers, including SCIP 7.0.2, can enumerate the set of feasible solutions $P^\mathcal{X}$ of (9) by adding a constraint handler whenever a new feasible integer variable assignment is found and continuing the solving process. We use this solution augmentation for our experiments on set covering and independent sets where the solver identified multiple optimal solutions in a short period of time.

# D  Background: Linear Programming

In this section, we briefly review some concepts from linear programming and duality.

**Definition 1 (Standard Form).** *A linear program $M^*$ of the form*

$$\min c^T x \quad \text{subject to } Ax = b, \ \underline{\pi} \le x \le \overline{\pi} \tag{10}$$

*is said to be in standard form with variable bounds. A basis for this program is a triplet $(\mathcal{L}, \mathcal{B}, \mathcal{U})$, where $x_{\mathcal{B}}$ are the basic variables and $x_{\mathcal{L}} = \underline{\pi}_{\mathcal{L}}$ and $x_{\mathcal{U}} = \overline{\pi}_{\mathcal{U}}$.*

**Definition 2 (Dual linear program).** *The dual linear program $M_D^*$ of the program given in 10 is*

$$\max_{y_b, y_{\underline{\pi}}, y_{\overline{\pi}}} y_b^{\mathsf{T}} b + y_{\underline{\pi}}^{\mathsf{T}} \underline{\pi} + y_{\overline{\pi}}^{\mathsf{T}} \overline{\pi} \quad \text{subject to } A^{\mathsf{T}} y_b + y_{\underline{\pi}} + y_{\overline{\pi}} = c, \ y_{\underline{\pi}} \ge 0, y_{\overline{\pi}} \le 0, \tag{11}$$

**Theorem 1 (Complementary Slackness).** *Let $x^*$ be a feasible solution to the linear program in (10) and let $y^* := (y_b^*, y_{\underline{\pi}}^*, y_{\overline{\pi}}^*)$ be a feasible solution to its corresponding dual linear program given in (11). Then, $x^*$ and $y^*$ are optimal solutions for the two respective problems if and only if*

$$(x^* - \underline{\pi}) \odot y_{\underline{\pi}}^* = 0 \tag{12}$$
$$(x^* - \overline{\pi}) \odot y_{\overline{\pi}}^* = 0 \tag{13}$$

*where $\odot$ denotes the Hadamard product. The additional conditions $y_b^* \odot (Ax - b) = 0$ and $\left(c - A^{\mathsf{T}} y_b - y_{\underline{\pi}} - y_{\overline{\pi}}\right) \odot x = 0$ are satisfied because both $x^*$ and $y^*$ are feasible.*

*Proof.* The conditions are derived from the definition of the dual program and strong duality in linear programming [see e.g., 10]. $\square$

# E  Additional experimental results

## E.1  Generalization to larger instances

For capacitated facility location, we evaluated the generalization performance of *L2Dive*. For our experiments in subsection 5.1, we trained and tested on the *small* instances from Gasse et al. [17] for each problem class. In this additional experiment, we tested the *L2Dive* model trained on the *small* capacitated facility location instances and evaluated it along with all standard divers on 100 *large* capacitated facility location test instances from [17]. We find that *L2Dive* gracefully generalizes to larger instances of the same problem class (Table 5). It achieves the lowest relative primal gap and outperforms all standard divers on the larger test instances. Overall, the relative primal gap tends to be larger which is expected as the larger instances tend to be more difficult to solve.

## E.2  Relative primal gap

In Table 6 we report the *relative* primal gap for the experiments in section 5.1 . The relative primal gap is computed as

$$\gamma_p'(\tilde{z}) = \frac{\gamma_p(\tilde{z})}{|z^\dagger|} \tag{14}$$

where $\gamma_p(\tilde{z})$ is as defined in equation (5). It is ill-defined, if the objective value is zero, but this was not the case for any instances considered in our experiments. As the instances within each of the four problem classes tend to have objective values of the same magnitude, the conclusions that can be drawn from the relative primal gap do not differ from those we can draw from the absolute primal gap in Table 1. We included the results here for completeness.

Table 5: *L2Dive* outperforms standard divers on *large* test instances from capacitated facility location, even when only trained on *small* instances. Average primal gap with standard error on test set.

|  | *small* | *large* |
|---|---|---|
| *L2Dive* | **0.89** (0.05) | **1.43** (0.11) |
| *Best heuristic* | *2.70* (0.04) | *2.25* (0.10) |
| *Coefficient* | 4.25 (0.05) | 5.35 (0.17) |
| *Distributional* | 4.24 (0.05) | 5.05 (0.17) |
| *Farkas* | *2.70* (0.04) | 3.16 (0.11) |
| *Fractional* | 5.91 (0.06) | 9.38 (0.20) |
| *Linesearch* | 5.78 (0.09) | 4.56 (0.19) |
| *Pseudocost* | 2.82 (0.06) | *2.25* (0.10) |
| *Vectorlength* | 4.68 (0.10) | 3.09 (0.19) |
| *Random* | 5.03 (0.08) | 4.10 (0.16) |
| *Lower* | 3.48 (0.05) | 4.01 (0.13) |
| *Upper* | 4.62 (0.08) | 3.17 (0.15) |

Table 6: We report mean and standard error of the relative primal gap in equation (14) for the experiments in subsection 5.1. The conclusions are the same as those drawn from Table 1.

|  | SET COVER | COMB. AUCTION | FAC. LOCATION | IND. SET |
|---|---|---|---|---|
| *L2Dive* | **2.86** (0.14) | **2.84** (0.09) | **0.89** (0.05) | **0.23** (0.02) |
| *Best heuristic* | *5.06* (0.16) | *3.27* (0.1) | *2.70* (0.04) | *0.82* (0.05) |
| *Coefficient* | 198.02 (2.23) | 8.56 (0.14) | 4.25 (0.05) | 10.91 (0.17) |
| *Distributional* | 209.29 (1.79) | 19.18 (0.15) | 4.24 (0.05) | 8.70 (0.11) |
| *Farkas* | 5.57 (0.17) | 6.08 (0.12) | *2.70* (0.04) | - |
| *Fractional* | 198.82 (2.26) | 8.57 (0.13) | 5.91 (0.06) | 10.27 (0.16) |
| *Linesearch* | 69.05 (1.29) | 5.96 (0.12) | 5.78 (0.09) | 3.42 (0.05) |
| *Pseudocost* | 10.44 (0.43) | *3.27* (0.1) | 2.82 (0.06) | 1.44 (0.08) |
| *Vectorlength* | *5.06* (0.16) | 10.65 (0.26) | 4.68 (0.10) | *0.82* (0.05) |
| *Random* | 22.07 (0.6) | 8.99 (0.16) | 5.03 (0.08) | 3.46 (0.11) |
| *Lower* | 153.59 (2.81) | 20.23 (0.14) | 3.48 (0.05) | 7.59 (0.20) |
| *Upper* | 12.69 (0.28) | 7.81 (0.14) | 4.62 (0.08) | 2.78 (0.09) |

### E.3 Execution Times

For completeness, we also report the execution times of all divers considered in the experiments in subsection 5.1 in Table 7. On the instances considered, diving tends to be rapid and differences in execution times are largely negligible (in contrast to differences in the quality of solutions found). Extremely short execution times typically result from diving being aborted early, e.g., because the LP was rendered infeasible.

## F Tuning the SCIP solvers for diving

We leveraged expert knowledge and used random search to optimize the use of diving heuristics in SCIP 7.0.2 for our baseline *Tuned*. The most important parameters to control the standard divers in SCIP are *freq* and *freqofs*. For each diving heuristic, these parameters control the depth at which the heuristic may be called or not called. By varying these parameters, diverse diving ensembles can be realized that call different heuristics at different stages of the branch and bound search. We randomly sample solver configurations by setting either $freq = -1$ (no diving), or $freq = \lfloor 0.5 \times freq_{\text{default}} \rfloor$ (double frequency) or $freq = freq_{\text{default}}$ (leave frequency at default) or $freq = \lfloor 2 \times freq_{\text{default}} \rfloor$ (halve frequency) with equal probability and setting either $freqofs = 0$ or $freqofs = freqofs_{\text{default}}$ with equal probability independently for each diving heuristic. We run the solver with the usual time limits for each configuration and each validation instance and pick the configuration with the lowest primal-dual

Table 7: We report mean and standard deviation of the execution time for the experiments in subsection 5.1. Diving on these instances is very fast and differences are mostly negligible. Extremely short execution times tend to indicate that diving was aborted early, e.g., because the LP was rendered infeasible.

|  | SET COVER | IND. SET | COMB. AUCTION | FAC. LOCATION |
|---|---|---|---|---|
| *L2Dive* | 0.45 (0.05) | 0.58 (0.15) | 0.38 (0.08) | 3.99 (0.76) |
| *Coefficient* | 0.28 (0.13) | 0.04 (0.01) | 0.02 (0.01) | 3.40 (0.61) |
| *Distributional* | 0.25 (0.11) | 0.04 (0.01) | 0.08 (0.02) | 3.41 (0.63) |
| *Farkas* | 0.05 (0.03) | - | 0.02 (0.01) | 3.66 (0.68) |
| *Fractional* | 0.29 (0.14) | 0.04 (0.01) | 0.02 (0.01) | 3.83 (0.60) |
| *Linesearch* | 0.10 (0.04) | 0.03 (0.01) | 0.03 (0.01) | 3.53 (0.79) |
| *Pseudocost* | 0.02 (0.01) | 0.04 (0.02) | 0.02 (0.00) | 2.77 (0.76) |
| *Vectorlength* | 0.02 (0.01) | 0.02 (0.01) | 0.05 (0.01) | 1.82 (0.59) |
| *Random* | 0.08 (0.04) | 0.25 (0.10) | 0.05 (0.01) | 4.42 (1.50) |
| *Lower* | 0.74 (0.43) | 0.66 (0.26) | 0.49 (0.07) | 4.11 (0.79) |
| *Upper* | 0.05 (0.02) | 0.17 (0.08) | 0.04 (0.01) | 2.79 (1.02) |

integral for server load balancing and with the lowest solving time for neural network verification. For server load balancing, since the original validation dataset was relatively small, we created a new validation dataset of 625 instances from the original training and validation sets. We optimized over 16 random configurations and thus used a budget of 10,000 solver calls for load balancing. For neural network verification, we used the original validation dataset of 505 validation instances. We considered six random configurations and thus used a budget of 3,030 solver calls which is slightly more than what *L2Dive* used for data collection (2903).

We did not tune any parameters to optimize the use of *L2Dive*, but this might improve performance.

