# OpenReview forum: "Learning To Dive In Branch And Bound"
_NeurIPS.cc/2023/Conference — NeurIPS 2023 poster_

### Official Review · Reviewer_cL66 · 2023-06-22

**Soundness:** 3 good
**Presentation:** 4 excellent
**Contribution:** 3 good
**Rating:** 5
**Confidence:** 5

**Summary:**

The authors propose L2Dive to learn dataset-specific diving heuristics with graph neural networks. Specifically, they employ generative models to predict variable assignments and leverage the duality of linear programs to make diving decisions based on the model’s predictions. Experiments show improved performance on both diving and branch-and-bound tasks.

**Strengths:**

1. Interesting research topic. The diving heuristic can be well formulated as a 0/1 prediction task. Thus, employing generative models for this task is meaningful and practical.
2. Novel algorithm design. The leverage of the duality property for variable selection is insightful, as it is not obvious to design such approach compared to the design of generative models.
3. Well written paper. The presentation is really clear, which makes the readers easy to follow.

**Weaknesses:**

1. Inefficient literature research. The main idea used in this topic is similar to that in [1] in the node selection task. Personally, I think the diving task and the node selection task on binary problems are really similar, and the GNN model proposed in [1] can be used in this task directly without much adaptations. Thus, the introduction and comparison to [1] is required.
2. (I am not sure for these results in my memory. So if I am wrong, please inform me the correct results.) Confusing results in Table 1. In my memory,  after presolve, SCIP can find optimal solutions easily at the root nodes on most instances from the four benchmarks the authors used. That is the reason why these four benchmarks are usually used for the variable selection task (dual task). Thus, I am confused that whether the results in Table 1 is convincing enough, as the benchmarks used here may be too easy for this task. (I am not sure about this comment)
3. Insufficient ablation study. The comparison bettween dual theory based variable selection and vanilla variable selection is missing. Thus, I am not sure whether the proposed dual based selection strategy is effective. The relationship between the complexity of the generative models and the final performance is also missing. The precision increases with more complex models, while the inference time also increases. Thus, there may be a trade-off for model design. I searched for the keyword "ablation" in both the main paper and the Appendix but have not found any results. For better completeness, it is better to include them in the *main paper*.
4. Insufficient comparative experiments. The generalization ability for this approach is missing. Thus, what is the performance of L2Dive on larger instances? Moreover, I observe that the results on the branch-and bound tasks are relatively limited. Thus, I also doubt about the overall performance on the B&B task as the generalization problem in this setting is usually more severe.

[1] Khalil, Elias B., Christopher Morris, and Andrea Lodi. "Mip-gnn: A data-driven framework for guiding combinatorial solvers." Proceedings of the AAAI Conference on Artificial Intelligence. Vol. 36. No. 9. 2022.

**Questions:**

1. The authors describe in line 257 that they "validated and tested on 500 instances". I am not sure whether they validate and test the model in the same dataset, which is discouraged in practice.
2. What is the standard deviation of the results in Table 1&2? As the solving time for different instances varies a lot, can you provide the geometric mean of these results as that in [2,3]?
3. I cannot find any code availability claims in both the paper and the appendix. I think this is not encouraged for an AI conference like NeurIPS. Can the codes of this paper be available once the paper is accept?

I would like to raise my score if  the authors reply well (especially question 3) to the above weaknesses and queations.

[2] Gupta, Prateek, et al. "Hybrid models for learning to branch." Advances in neural information processing systems 33 (2020): 18087-18097.
[3] Achterberg, Tobias. Constraint integer programming. Diss. 2007.

**Limitations:**

See weakness above.

---

> ### Author Rebuttal · Authors · 2023-08-10
>
> # Thank you. We made revisions based on your feedback.
>
> **Thank you very much for your thoughtful review. We have made revisions to our paper based on your feedback and address your concerns in more detail below.**
>
> &nbsp;
> ## We performed additional ablations to isolate the effects of dual reasoning.
> >*“Insufficient ablation study. The comparison bettween dual theory based variable selection and vanilla variable selection is missing.*
>
> Thank you for this suggestion! We fully agree that this ablation is a good idea! And it was also proposed by Reviewer gbBh. We have carried out an ablation study on capacitated facility location, where we evaluated
>
> - L2Dive with $s_j = q_\theta(\hat{x}_j)$, i.e. only using model confidence, not using dual reasoning.
> - L2Dive with $s_j = \text{Uniform}(0, 1)$, i.e. a random variable selection, not using model confidence, not using dual reasoning.
>
> The results are shown in the attachment. Even with a random scoring rule, L2Dive outperforms the best heuristics. However, the results significantly improve when using model confidence, and they slightly improve when additionally using dual reasoning. Since using dual reasoning comes at negligible additional costs, because the dual values are readily available in the solver, we choose to use dual reasoning. We have added these results to the main text along with a short discussion.
>
> &nbsp;
> ## Khalil et al. is relevant and related, but different.
> >*“main idea [...] is similar to that in [1] in the node selection task [...] introduction and comparison is required.”*
>
> Thank you for bringing Khalil et al. [1] to our attention. This is a cool paper! Like us they leverage supervised learning on solutions of MILP to train a GNN for predicting variable assignments, (albeit both our model architecture and objective differ from theirs). But most importantly, as you note, they propose a heuristic rule to perform node selection from the model prediction, while we consider diving. One of our key contributions is to propose a method for using a generative model for diving (section 3.2). Our approach offers several benefits, e.g., (a) depth-first search for node selection is known to be problematic, because node selection needs to balance the primal and dual objective, while diving is solely concerned with the primal objective (Achterberg, page 73), (b) diving can leverage the fast LP solver rather than fully (and time-consumingly) resolving each node in the search tree, (c) since diving is conducted in probing mode, a “bad” dive leading to many open nodes can simply be aborted without impacting the solver state, while bad node selection decisions must be undone via a solver restart.
>
> We think this is a valuable discussion. Therefore, we have added a reference to Khalil et al [1] in the main text, and clearly highlight similarities and differences.
>
> &nbsp;
> ## Problem instances are large-scale and experimental protocol is thorough.
> >*“results on the branch-and bound tasks are relatively limited. [...] performance of L2Dive on larger instances? ”*
>
> We politely disagree. The instances we consider in our experiments in Section 5.2 are large-scale. For example, the load balancing instances contain 61,000 variables and 64,410 constraints per instance prior to pre-solve and more than 4,000 variables and 4,000 constraints after pre-solve. Similarly, the neural network verification instances are of the same order of magnitude.
> This is comparable or larger to the “hard” instances considered by Gasse et al. (2019). In addition, our instances are harder to solve. Not a single load balancing instance can be solved with the SCIP solver within 60 minutes, while the results reported in Gasse et al. suggest that their instances can be solved on average in 2 minutes (set cover) to 35 minutes (independent set).
> Moreover, our experiments on branch and bound are extensive and thorough. Notably, we compare against (and outperform) a strong “tuned” solver baseline, something that is arguably important fto benchmark machine learning methods for branch and bound against, but rarely done (with some notable exceptions in [Nair et al. (2020),  Sonnerat et al. (2021), Chmiela et al. (2021) ]. We establish a thorough protocol (Appendix E) for evaluation where our baseline receives the same budget of solver calls for tuning as L2Dive expended for data collection.
> Overall, we consider a total of six datasets.
>
> &nbsp;
> ## Experiments in 5.1 are for comparing diving heuristics
> >*“benchmarks [used in 5.1] too easy for this task?”*
>
> Thank you for raising this point. Our goal for the experiments in Section 5.1 was to study the diving performance of L2Dive and compare it to existing diving heuristics. While it is true that these instances can be solved relatively quickly with a full branch and bound solver, our results suggest that they are challenging for diving heuristics (larger primal gaps), and thus these instances are appropriate and provide a useful testbed. We chose these instances, because they have been adopted by prior work in machine learning for branch and bound and represent a diverse set of combinatorial problems.
>
> &nbsp;
> ## Validation and test set are different!
> > *“validate and test the model in the same dataset?”*
>
> Absolutely not! We apologize for the confusion. Validation and test set are different! Please see response to qRaC
>
>
> &nbsp;
> ## Code will be made publicly available upon acceptance.
> > *“Can the codes of this paper be available once the paper is accepted?”*
>
> Absolutely! We will make all experimental code publicly available upon acceptance. In addition, we are in discussions with the SCIP/ PySCIPOpt development team to contribute a new diving plug-in available to the PySCIPOpt library. We hope our contributions will facilitate further research into machine learning for branch and bound generally, and diving heuristics specifically.

---

> > ### Comment · Reviewer_cL66 · 2023-08-17
> >
> > Thanks you for the response. Based on your additional experimental results and the reviews from other reviewers, I raised my score to 5 (borderline accept).
> >
> > However, there are still several questions I am concerned about:
> >
> > - Related work about node selection. *depth-first search for node selection is known to be problematic, because node selection needs to balance the primal and dual objective, while diving is solely concerned with the primal objective*. In fact, I found most existing research on node selection only focus on finding a good primal solution. Perhaps considering primal and dual objectives together could be extremely challenging.
> >
> > - Benchmark selection. I ran SCIP again on SetCover and IndSet to check my previous reviews. As I expected, SCIP achieves best primal solution on most instances *at the root node*. which suggests that these problems are too simple for primal task evaluation. The four benchmarks are widely used for dual tasks, but are not the best choices for primal task.
> >
> > Anyway, I still agree L2Dive is valuable for the community of combinatorial optimization, as currently, the research (*with open-sourced codes*) is relatively limited.

---

> > > ### Author Response · Authors · 2023-08-17
> > > **Thank you for your feedback!**
> > >
> > > > *I still agree L2Dive is valuable for the community [...]*
> > >
> > > Thank you! We are glad you approve of our work! We briefly respond to your two outstanding concerns below.
> > >
> > > ---
> > >
> > > > *[...] research on node selection only focus on finding a good primal solution*
> > >
> > > Thank you! We were referring to Achterberg (Constraint Integer Programming, 2009) in our previous response, and would like to share: *The selection of the subproblem that should be processed next has two usually opposing goals within the MIP branch-and-bound search: 1. Finding good feasible MIP solutions to improve the primal (upper) bound, which helps to prune the search tree by bounding, and 2. Improving the global dual bound (Page 73).*
> > >
> > > Achterberg discusses depth-first search for node selection in section 6.1 and discusses *best-first search* in section 6.2 which *”aims at improving the global dualbound as fast as possible"* (Page 74). We agree that balancing primal and dual objectives for node selection is extremely challenging and subject to ongoing research. In our work, we focus on the task of finding good feasible solutions via the diving subroutine.
> > >
> > > > *SCIP achieves best primal solution on most instances at the root node [for set cover and independent set]*
> > >
> > > We would like to clarify that the discrepancy is explained by the fact that we disable separation and other heuristics (cf Section 5), as our interest is in measuring diving performance without confounding with other parts of the solver. Our results (Table 1) indicate that these instances are challenging for existing diving heuristics in this setting. In particular, switching off separation can make the instances significantly harder for diving heuristics to successfully solve. We would like to affirm that in section 5.1 our main interest is comparison to existing diving heuristics, while in section 5.2, we consider diving in a full-solver environment to show improvements in overall solver performance with *L2Dive* on *other* large-scale instances. We also highlight the additional ablation study we performed for the experiments in section 5.1, where we show that *L2Dive* generalizes gracefully to larger instances (cf. Comment to Reviewer gbBh). These instances tend to be harder for SCIP, but the insights they afford in comparing diving heuristics corroborated our earlier findings on the smaller-sized instances.

---

### Official Review · Reviewer_GD8n · 2023-07-05

**Soundness:** 4 excellent
**Presentation:** 3 good
**Contribution:** 4 excellent
**Rating:** 5
**Confidence:** 3

**Summary:**

This paper introduces a new framework to improve B&B MIP solvers by neural networks. This paper proposes a neural network-based primal heuristic, namely L2Dive. The authors implement L2Dive with SCIP and conduct extensive experiments on several datasets.


---------------------

Post-rebuttal: Thanks a lot for the response. I feel positive about this paper and will vote for accept.

**Strengths:**

* This paper shows the feasibility of using neural networks to replace & improve existing primal heuristics.
* The design details of L2Dive seem technically sound.
* The experiments are extensive and convincing.

**Weaknesses:**

* The authors should provide the timing statistics of different primal heuristics in Table 1.
* As I know, Nair et al. [32] proposed "neural diving" and the conceptual idea is very similar to the contribution of this paper. The authors should make more efforts to address the original technical contributions compared to [32] and try to compare with [32] in experiments.
* Some details of the approach remain unclear to me:
    * The neural network is trained with high-quality solutions, while the output of the neural network represents the score of whether a variable should be dived. How do you connect these two? What is exactly the loss function during training?
    * How many times is the neural network called when running Algorithm 1?

**Questions:**

* Why different measurements are considered for different tasks in Table 2?

---

> ### Author Rebuttal · Authors · 2023-08-10
>
> # Thank you for your review.
>
> **Thank you very much for your feedback. We address your concerns in more detail below.**
>
> &nbsp;
> ## Other primal heuristics are complementary to diving, no heuristic rules them all!
> >*“The authors should make more efforts to address the original technical contributions compared to Nair et al. [32] and try to compare with [32] in experiments.*
>
> We view Nair et al. [32] as complementary, because it is a primal, but not a diving heuristic. In our experiments in section 5.2, we deploy L2Dive within branch and bound, and our model is called by the solver along with many other subroutines (including other primal heuristics) and the method by Nair et al. [32] could also be used in addition to improve performance further. Intuitively, we expect L2Dive to be particularly effective for instances where the solutions of linear programs can be rendered integral easily, and we expect Nair et al. [32] to be particularly effective on instances, where the overhead of solving sub-MILPs in place of linear programs tends to be small.
>
> Overall, the goal of our work is not to demonstrate the superiority of diving heuristics over other primal heuristics, but to improve on existing diving heuristics with machine learning. We believe diving heuristics play an important role in branch and bound, but so do other primal heuristics, and which method is more appropriate will depend on the particular problem instances under consideration. This is the reason why we compare primarily against other diving heuristics.
>
> We think this is an important discussion, and we have expanded our related work section to include it.
>
>
> &nbsp;
> ## Variable selection is based on model confidence and dual reasoning. Model is only called once.
> >*"The neural network is trained with high-quality solutions, while the output of the neural network represents the score of whether a variable should be dived. How do you connect these two? What is exactly the loss function during training?"*
>
> You are spot on! Our objective is to minimize the KL divergence in equation (7) and at training time our loss function is a mini-batched average of this KL divergence. Our model is only learnt from good feasible solutions, there is no direct supervision on diving or variable selection. While this choice offers several advantages, e.g., much cheaper to collect training data, the model is trained to only predict correct variable assignments, but not what variables to choose to fix for diving.
>
> To bridge this gap, we suggest the rule in equation (8) that balances the model confidence in a prediction with duality theory to encourage shorter dives. Intuitively, the dual reasoning rule is justified if the prediction is correct. The better the prediction, the more confident the model, the better the dual reasoning rule is expected to work. Our empirical results along with the additional ablations we performed support our choice.
>
> >*“How many times is the neural network called when running Algorithm 1?”*
>
> Thank you, this is an important question! For any dive, we only call the model once at the beginning of the dive. While it’d be possible to call the model more frequently to potentially improve predictions, this keeps the in-service overhead of our method low and simplifies considerations for data collection.
>
> &nbsp;
> ## Load balancing instances are hard, L2Dive is fast.
> >*“Why are different measurements considered for different tasks in Table 2?”*
>
> Table 2 reports the primal-dual integral for instances from load balancing, but solving time for instances from neural network verification. The goal of Table 2 is to report the overall solver performance for different methods on each task (dataset). Overall solver performance is naturally measured by solving time, i.e., the total time it takes to completely solve an instance (primal-dual gap is zero). However, the load balancing instances are so difficult to solve, that they cannot be solved in a reasonable amount of time. For example, not a single instance can be solved using SCIP (Default) within 60 minutes. Therefore, we set a time limit of 900 seconds and report the primal-dual integral instead. The primal-dual integral is an accepted measure of solver performance (Achterberg et al.).
>
> Thank you for raising this point. We now highlight the reason for reporting the primal-dual integral for load balancing in Table 2 more prominently.
>
> >*The authors should provide the timing statistics of different primal heuristics in Table 1.*
>
> We are happy to report the overall execution times for all divers in the attachment and will include them in the appendix, but did not do so initially, because they are largely negligible. On set cover, independent set and combinatorial auctions, all divers take less than a second to run, while on facility location all divers tend to run in between three to four seconds. For all instances, execution time is mostly correlated with the depth of the dive. A method that renders an instance quickly infeasible and thus aborts the dive (unsuccessfuly) will have a small execution time, while a method that dives deep will have a longer execution time, but not necessarily a good solution (for example Lower on set cover runs 50% longer than L2Dive). L2Dive naturally incurs a small overhead for calling the neural network, but it also consistently finds better solutions (Table 1 in main text).

---

> > ### Comment · Reviewer_GD8n · 2023-08-18
> >
> > Thanks a lot for the response. I feel positive for this paper and will vote for accept.

---

### Official Review · Reviewer_qRaC · 2023-07-05

**Soundness:** 4 excellent
**Presentation:** 4 excellent
**Contribution:** 3 good
**Rating:** 7
**Confidence:** 4

**Summary:**

This paper presents a technique for heuristically generating good feasible solutions for mixed-integer programming problems by leveraging known, good feasible solutions for related problem instances. The technique is a "diving" heuristic, which subsequently fixes subsets of the integer decision variables, and uses a generative model (and a bit of LP duality) to determine the subsets. A computational study suggests that their technique outperforms the diving heuristics contained in the SCIP solver.

**Strengths:**

This paper falls squarely in a line of research (learning for optimization) that is of substantial interest to the NeurIPS and mathematical optimization communities. The ideas are interesting and nontrivial (the application of LP duality is neat and clean, if not especially deep). The contributions are, to my reading, quite solid, though the general area in which they are operating has been studied for some years. The paper is very well-written and, to my understanding, original, though there are a handful of closely related papers that the authors situate themselves against explicitly.

**Weaknesses:**

As the authors themselves recognize (to their credit), the computational baseline against SCIP's existing diving heuristics may be a weak one. But, as SCIP is a best-of-breed open MIP solver and this study requires tight integration into the solver, I do not see this as something that can be held against this paper.

**Questions:**

* L49: Unfinished sentence.
* L149: What are "trivial solutions"?
* Proposition 1: The paper should be self-contained, and not require the appendix to comprehend. I'd suggest listing the dual LP in the text, and perhaps move some of Section 2.2 to the appendix if space is needed.
* Citation [17] has broken typesetting.
* L257: Can the authors please confirm that a _different_ set of 500 instances are used for validation and testing?
* Section 5.1 could be further strengthened by comparing against other, non-diving, heuristics in SCIP.

**Limitations:**

Yes.

---

> ### Author Rebuttal · Authors · 2023-08-10
>
> # Thank you for your positive reception of our work.
>
> **We are happy to integrate the dual LP into the main text. We have fixed the typos you found in the main text. We address your outstanding questions below.**
>
> >*“Please confirm that a different set of 500 instances are used for validation and testing?”*
>
> Absolutely! We apologize for the confusion. They are different! Specifically, for each dataset considered in Table 1, we  generated 2000 (iid) instances in total, and used 1000 of these instances for training, 500 of these instances for validation and the remaining 500 of these instances for testing. This question was also raised by Reviewer cl66  and we have edited the main text to make this crystal clear.
>
>
> >*“Existing diving heuristics may be weak, [... cannot] be held against this paper”*
>
> Thank you! We agree and would like to confirm that unfortunately Gurobi’s diving code is closed-source to date and thus does not facilitate any experiments with *L2Dive*. We also highlight that we compare against a *tuned* diving ensemble in section 5.2 which significantly improves over SCIP’s default setting and is a strong baseline for direct comparison.
>
> >*“What are trivial solutions (Line 149)?”*
>
> We were originally referring to the trivial solutions that SCIP tests in the beginning of a solve, e.g., setting all variables to zero. However, we believe that “poor feasible solutions” is much clearer in this context. Thank you for flagging this, we have changed the main text!

---

> > ### Comment · Area_Chair_UPkK · 2023-08-18
> > **Thanks**
> >
> > Thank you for this feedback authors. This will be taken into account.

---

### Official Review · Reviewer_gbBh · 2023-07-06

**Soundness:** 3 good
**Presentation:** 2 fair
**Contribution:** 3 good
**Rating:** 7
**Confidence:** 4

**Summary:**

The paper develops a learning strategy to enhance variable selection in primal diving heuristics. More specifically, such a methodology relies on a generative model based on graph neural networks to predict the likelihood of variables assuming specific values. The predictions of the mean values are then integrated with dual reasoning (i.e., determining whether tightening would lead to an optimal linear programming solution) to decide the next variable to dive into. Numerical results evaluate the impact of the heuristic on a class of four combinatorial optimization problems, as well as its impact on a complete branching method for load balancing and neural network verification instances.

**Strengths:**

+ Well-motivated application.
+ Model is nicely designed.

The paper contributes to the research stream of machine-learning methods incorporated into combinatorial methods. I found that the generative model is compelling and fits well within diving heuristics, and it could as well be applied to other similar primal heuristics. The solution augmentation is indeed a challenge for training, but I appreciate the discussion and I also found the counting strategy quite novel (albeit possibly expensive?). I imagine that could be considered in other contexts as well, especially when providing sufficiently diverse solutions.

**Weaknesses:**

- Numerical results lack detail and more thorough analysis.
- Presentation needs a few revisions.

My primary concern is that results are not examined in detail; the proposed L2Dive technique completely dominates the other heuristics on the primal gap (Table 1) and has very small standard errors; for n=1,000 instances, this would indicate that the variance is quite small across a large instance class. This is somewhat rare in primal heuristic development and for such a variety of instances, and it is unclear from the text why this is happening. More specifically:

(a) The text does not detail how instances of the four combinatorial optimization problems are selected. In particular, it refers to reference [17] for the benchmark tested, but in that paper they utilize multiple combinations of "easy", "medium," and "hard" cases for each class; I am not sure which ones are considered in the paper. The training set choice is not clear as well. For instance, given that Erdos-Rényi instances are so particularly structured, are the independent set results really representative in practice in this particular context? It could be more beneficial to consider, e.g., DIMACS cases, which present a more diverse structure.

(b) The paper only reports the non-normalized $\textit{absolute}$ primal gap, which is the difference between the primal solution value and an optimistic bound. In this case, it is difficult to assess how significant the improvements are; for example, "222" and "256" could effectively have a small difference in terms of the primal-dual gap in combinatorial auctions if the bound is large.

(c) One important missing analysis is the relevance between the model $q_{\theta}$ and the prioritization given by Proposition 1. First, how would LP2Dive behave without the indicator term from Proposition 1? Or, in other words, is the learning effective? Second, what would happen to the other diving heuristics (coefficient, Farkas etc.) if variables are first prioritized by Proposition 1?

I also believe the presentation needs to be revised. In particular:

- There are several typos and incomplete sentences. For example, "verificationw" (l.13), missing "in Figure 1" (l.50), adjectives miss hyphen (branch and bound tree --> branch-and-bound tree), and errors in references (e.g., in [17])
- Table legends should start with what the table represents and not with what their aims, as that is a bit confusing.
- Figure 1 is a bit informal and not needed, as the overall sequential process is relatively easy to understand.
- One suggestion is that $\pi$ is typically used as dual variable as opposed to a bound. Perhaps $\ell$ and $u$ could be clearer?

** Updated score after thorough discussion with the authors.





**Questions:**

1. From my understanding, if variables are discrete then $q_{\theta}$ provides a vector of means for a Bernoulli sequence. How would $j^*$ be calculated in this case?

2. Could you the authors kindly comment on (a), describing how instances for both the training and evaluation sets are picked?

3. Have the authors performed experiments with/without Proposition 1?

4. Would it be possible to augment any of the diving heuristics with Proposition 1 to evaluate if they would improve performance?

5. Could the authors also include table with the primal-dual gap?

**Limitations:**

Limitations were properly addressed.

---

> ### Author Rebuttal · Authors · 2023-08-10
>
> # Thank you. We made revisions based on your feedback.
>
> **Thank you very much for your thoughtful review. We have made revisions to our paper based on your feedback and address your concerns in more detail below.**
>
> &nbsp;
> ## We performed additional ablations to isolate the effects of dual reasoning.
> >*“experiments with/without Proposition 1? [...] L2Dive without the indicator term in Proposition 1?  [...] is the learning effective?”*
>
> Thank you for this suggestion! We fully agree that this ablation is a good idea! And it was also proposed by Reviewer cl66. We have carried out an ablation study on capacitated facility location, where we evaluated
>
> - L2Dive with $s_j = q_\theta(\hat{x}_j)$, i.e. only using model confidence, not using dual reasoning.
> - L2Dive with $s_j = \text{Uniform}(0, 1)$, i.e. a random variable selection, not using model confidence, not using dual reasoning.
>
> The results are shown in the attachment. Even with a random scoring rule, L2Dive outperforms the best heuristics in terms of primal gap. However, the results significantly improve when using model confidence, and they tend to slightly improve when additionally using dual reasoning. Since using dual reasoning comes at negligible additional costs, because the dual values are readily available in the solver, we choose to use dual reasoning. We have added these results to the main text along with a short discussion.
>
> >*"Would it be possible to augment any of the diving heuristics with Proposition 1 to evaluate if they would improve performance?"*
>
> We agree that this is an interesting avenue for future work in operations research, but we are primarily concerned with developing machine learning methods. Since we rely on the plug-in to call all standard divers in SCIP  (Section 3.3), using a dual reasoning rule for existing diving heuristics will require editing their source in SCIP directly and is out of scope for this work. However, we will release our code publicly upon acceptance and hope that this will facilitate research in this direction.
>
> &nbsp;
> ## Experiments in 5.1 are intended to assess primal performance.
> >*““only reports non-normalized absolute primal gap […], what about primal-dual gap?“*
>
> We report the primal gap in Table 1, because this is the bound that any diving heuristic will effectively improve. We choose to report the absolute primal gap, because it is conceptually easier. We were initially concerned about using the relative gap as defined in SCIP, because it is ill-defined if the primalbound and objective value have opposite signs and it may choose to normalize the absolute gap using either the primalbound or the objective value which can hinder comparisons between divers.
>
> However, we are happy to report the relative primal gap where we always normalize by the absolute objective value (we found that for all instances, primalbounds and objective values are of the same sign) in the attachment and now include these additional results in the appendix.
>
> The experiments in section 5.1 were designed to study the diving performance of L2Dive and compare it against existing diving heuristics without confounding the results with other subroutines of the solver. Therefore, we did not branch and disabled cutting planes and other primal heuristics. Hence, we do not compute the dualbound or the primal-dual gap in these experiments, they are naturally weak. We focus on overall solver performance (primal-dual integral, solving time) in our experiments in section 5.2.
>
> &nbsp;
> ## Data is appropriate for our experimental goals.
> >*““How instances of the four combinatorial optimization problems are selected?” Are independent set results really representative […]?“*
>
> We use the “easy” (i.e., smaller-sized) instances for our experiments. These instances were randomly generated with exactly the same procedure as described by Gasse et al using the generators made available in Ecole [Prouvost et al., 2020]. We now include the explicit reference to the Ecole library in the main text.
>
> We chose these instances, because they collectively represent a diverse set of combinatorial problems and they have been adopted by previous work [e.g., Gasse et al. (2019), Gupta et al. (2020), Scavuzzo et al. (2022) among others]. We believe there is value in using the same problem benchmark. We chose the “easy” instances, because they allowed us to test many methods at scale with relatively modest computational resources. In addition, our results indicate that they are “sufficiently hard” for existing diving heuristics (larger average primal gaps). Thus, they provide a useful test bed. In section 5.2, we focus on overall solver performance in real-world application and use harder and larger instances.
>
>
> &nbsp;
> ## Table 1 reports the standard error of the mean, not the standard deviation of the sample.
> >*“The primal gap (Table 1) has very small standard errors. [...] for n=1,000 instances, this would indicate that the variance is quite small across a large instance class.“*
>
> We apologize for the confusion. Table 1 reports the standard error of the mean (SEM), not the standard deviation of the sample (all n observations). Specifically, the standard error of the mean is defined as
>
> $$
> \text{SEM} = \frac{s}{\sqrt{n}}
> $$
>
> where $s$ is the standard deviation of the sample computed for $n=500$ observations. Naturally, as the sample size $n$ increases, the standard error of the mean is reduced, as the estimate becomes more accurate. We choose to report the standard error of the mean, because statistical significance can be more easily gauged (e.g., via t-test) from it than from the standard deviation of the sample. We find that the standard deviation on all four datasets is of the same order as the standard deviation of the objective value and find no anomaly. We are happy to report the standard deviation in the main text or the appendix if others find it helpful.

---

> > ### Comment · Reviewer_gbBh · 2023-08-13
> >
> > Thank you for the detailed responses.
> >
> > I believe the results concerning the optimality gaps are more convincing and place the approach in a better light. It is difficult to grasp what absolute primal values mean as problem classes are quite diverse.
> >
> > The ablation is also quite intriguing. What made model confidence so impactful for the facility location instances?
> >
> > Further, what happens when diving is applied to the "difficult" instances? Even if the benchmark is appropriate as a means to test baseline methods, I would imagine that the contribution could be more impactful if showing how ML-based help address those challenging problems.
> >
> > Finally, I did understand what the standard error was; my comment is that the variation is small because $\sqrt{500}$ is not large relative to the primal gaps reported. My question remains on why variance was not high, as typically seen in more heuristic approaches.

---

> > > ### Author Response · Authors · 2023-08-16
> > > **Thank you for your feedback! New ablation!**
> > >
> > > Thank you for your feedback! Please see our individual responses below
> > >
> > > > *the results concerning the optimality gaps are more convincing*
> > >
> > > Thank you! We do agree and we are happy to report results for the optimality gaps rather than the absolute primal gap in the final manuscript!
> > >
> > > > ”ablation is also quite intriguing. What made model confidence so impactful [...]?
> > >
> > > Thank you for your interest and for having suggested this ablation! We observed the predictions in which the model was highly confident (and which were thus chosen when model confidence was used) tended to have a higher accuracy (in predicting the optimal solution) than those where the model was less confident. When model confidence was not used, it was more common to fix variables to values that rendered the problem infeasible earlier in the dive, and thus led to poor solutions. We are happy to add this discussion to the results of the ablation.
> > >
> > > > *“What happens when diving is applied to the "difficult" instances?” [...] showing how ML-based help address those challenging problems*
> > >
> > > As you recognized, the main objective of our experiments in section 5.1 was to test many methods at scale with relatively modest computational resources (computing primal gaps requires solving each test instance to optimality). We consider large-sized instances in a full-solver environment in section 5.2 where we report total improvements in overall solver performance with *L2Dive* which is arguably our most important experimental contribution.
> > >
> > > However, we did not want to shy away from your question! We took the opportunity to evaluate L2Dive and all baselines divers on 100 instances of the “hard” capacitated facility location instances, which are the largest-sized instances of the four classes. However, we did not train our model on the “hard” instances, instead we evaluated the model that was previously trained on the “easy” instances. This is in line with the suggestion of Reviewer cl66 who proposed a “generalization” ablation. The results are displayed below:
> > >
> > > | Diver | Relative Primal Gap (standard error) |
> > > |:----------|----------:|
> > > | *L2Dive* | **1.43 (0.11)** |
> > > |   Coefficient | 5.35  (0.17) |
> > > | Distribution | 5.05 (0.17) |
> > > | Farkas | 3.16 (0.11) |
> > > | Fractional | 9.38 (0.20) |
> > > | Linesearch | 4.56 (0.19)|
> > > | Pseudocost | 2.25 (0.10) |
> > > | Vectorlength | 3.09 (0.19) |
> > > | Random | 4.10 (0.16) |
> > > | Lower | 4.01 (0.13) |
> > > | Upper | 3.17 (0.15) |
> > >
> > > First, we observe that the optimality gap (relative primal gap) tends to be larger or similar for most heuristics which is expected as the instances are more difficult. Second, we observe that L2Dive still outperforms standard diving heuristics on the larger instances which suggests that the model is able to generalize to larger instances, even though it was only trained on the smaller instances.
> > >
> > > We think these results are interesting and strengthen our work and we are happy to include them in the final manuscript.
> > >
> > > > “why variance was not high, as typically seen in more heuristic approaches?”
> > >
> > > Thank you for your clarification!
> > > - First, we observe that the standard deviation for both L2Dive and other primal heuristics approaches is comparable on all four datasets (Table 5.1). It tends to be slightly larger for heuristics that struggle to find good solutions on average (e.g., Coefficient on Set Cover). We observed that this tends to be, because these heuristics find poor solutions on most instances, and good solutions only a few instances, which tends to increase variance, compared to better performing heuristics and L2Dive that consistently find good solutions.
> > > - Secondly, we generally expect the standard deviation of the primal gaps to be higher on datasets where the structural differences between instances tend to be larger and their objective values may fluctuate more widely (possibly by orders of magnitude). The standard deviation of the objective value on our test sets are 26, 323, 683, 4 for set cover, combinatorial auctions, facility location and independent set respectively, and thus corroborate this intuition. To generate these instances, we used the standard generators from Ecole library with default parameters as reported in Gasse et al. (2019) which was used in other work.
> > > - Finally, we’d like to affirm that the focus of our work is to learn problem-specific diving heuristics for particular applications. Hence, instances that share structural commonality and may feature smaller standard deviation in their objective value do not seem inappropriate. However, we are aware that there are limitations in learning diving heuristics for specific applications (cf. Broader Impact) and there is value in studying the transfer of models and the development of general-purpose models that perform well on diverse problem collections. This is a natural direction to explore in the future, for this and for other work at the intersection of machine learning and integer programming.

---

> > > > ### Comment · Reviewer_gbBh · 2023-08-17
> > > >
> > > > Thank you for the excellent response and for your effort in addressing all my comments. I would just recommend finding a way to incorporate those comments and observations, even if in the appendix. I am updating my score accordingly.

---

> > > > > ### Author Response · Authors · 2023-08-17
> > > > > **Thank you!**
> > > > >
> > > > > Thank you for the productive discussion!  We are happy that we were able to address all your concerns and are grateful for your engagement and your help in strengthening our work. We confirm that we will include the additional findings and commentary in the final manuscript!

---

### Author Rebuttal · Authors · 2023-08-10

## Thank you for your thoughtful feedback!

**We have incorporated the feedback of the reviewers and made the following revisions to our draft:**

- We present an additional ablation study (see attachment!) on capacitated facility location to study L2Dive’s variable selection (dual reasoning and model confidence) in more detail. We propose to add it to the main text along with a short discussion and interpretation of the results.
- We report additional metrics (e.g., relative gap, execution time, see attachment!) as suggested and propose to add them to the main text/ appendix.
- We added discussions of relevant related work as suggested in the main text.
- Small edits (typos, clarifications, etc.), thank you all for flagging any you found!

Please see individual responses for more detail below.

---

### Decision · Program_Chairs · 2023-09-21

**Decision:**

Accept (poster)

**Comment:**

All reviewers agreed this paper should be accepted: it addresses an important problem, the algorithm is well-designed and novel, and the paper is clearly written. A clear accept. Authors: you've already indicated that you've updated the submission to respond to reviewer changes, if you could double check their comments for any recommendation you may have missed on accident that would be great! The paper will make a great contribution to the conference!